# Towards Non-Targeted Screening of Lipid Biomarkers for Improved Equine Anti-Doping

**DOI:** 10.3390/molecules28010312

**Published:** 2022-12-30

**Authors:** Kathy Tou, Adam Cawley, Christopher Bowen, David P. Bishop, Shanlin Fu

**Affiliations:** 1Centre for Forensic Science, University of Technology Sydney, Sydney, NSW 2007, Australia; 2Australian Racing Forensic Laboratory, Racing NSW, Sydney, NSW 2000, Australia; 3Mass Spectrometry Business Unit, Shimadzu Scientific Instruments (Australasia), Sydney, NSW 2116, Australia; 4Hyphenated Mass Spectrometry Laboratory (HyMAS), University of Technology, Sydney, NSW 2007, Australia

**Keywords:** lipidomics, review, analytical, corticosteroids, NSAIDs

## Abstract

The current approach to equine anti-doping is focused on the targeted detection of prohibited substances. However, as new substances are rapidly being developed, the need for complimentary methods for monitoring is crucial to ensure the integrity of the racing industry is upheld. Lipidomics is a growing field involved in the characterisation of lipids, their function and metabolism in a biological system. Different lipids have various biological effects throughout the equine system including platelet aggregation and inflammation. A certain class of lipids that are being reviewed are the eicosanoids (inflammatory markers). The use of eicosanoids as a complementary method for monitoring has become increasingly popular with various studies completed to highlight their potential. Studies including various corticosteroids, non-steroidal anti-inflammatories and cannabidiol have been reviewed to highlight the progress lipidomics has had in contributing to the equine anti-doping industry. This review has explored the techniques used to prepare and analyse samples for lipidomic investigations in addition to the statistical analysis and potential for lipidomics to be used for a longitudinal assessment in the equine anti-doping industry.

## 1. Biomarkers for Equine Anti-Doping

The current approach to equine anti-doping is focused on the targeted detection of prohibited substances [1]. However, as new substances are rapidly being developed, the need for complimentary methods of monitoring is important to ensure the integrity of the racing industry is upheld [1]. The use of biomarkers for the detection of doping abuse is a significant advancement for sports anti-doping. Teale et al. [2] define biomarkers as an “individual biological parameter or substance (metabolite, protein or transcript); the concentration of which is indicative of the use or abuse of a drug or therapy”. With the discovery of novel biomarkers for detecting doping abuse, the potential exists for a larger number of drugs to be indirectly detected and over longer periods of time. However, with indirect detection, there is the possibility of the method not being specific and the increased likelihood of inconsistent results [3]. This review will focus on the use of lipidomics for indirect detection of substances in equine racing, contributing to an intelligence-based anti-doping strategy [4].

## 2. Current Challenges for Targeted Screening

Doping practices are increasingly sophisticated for all sports with testing laboratories consistently improving their workflows to identify the ever-growing list of prohibited substances. Current detection methods are, however, by nature limited in scope to a defined number of substances and the applicability of the analytical technique used [5]. Maintaining a contemporary scope of testing makes direct detection continually difficult due to availability of reference materials. An “omics” approach may provide an alternative to direct detection of doping [5,6]. The use of metabolomics has been utilised in many different laboratories to measure metabolites at low levels relative to time-related biological responses of a drug administration [3,5,6]. This provides a framework for non-targeted detection, particularly for drugs that have a short half-life but long lasting effect on any individual system [5].

## 3. Lipids

Lipidomics is a growing field involved in the characterisation of lipids, their function and metabolism in a biological system [7,8]. Lipids are non-polar molecules with a diverse chemistry and functionality [9,10]. In conjunction with carbohydrates, lipids are the main energy source for equine striated muscles [11]. There are a number of different classes of lipids including monounsaturated fatty acids (MUFAs) and polyunsaturated fatty acids (PUFAs) [12,13]. MUFAs are lipids that have a single double bond present in the compound and usually only exist in seeds or marine organisms, however, are naturally rare [12]. PUFAs comparatively contain more than one double bond, are more commonly found [12] and have various biological effects including platelet aggregation and inflammation [14]. In animals, common PUFAs include arachidonic acid (AA), eicosapentaenoic acid (EPA) and docosahexaenoic acid (DHA) [12]. The most relevant and important oxygenated products for the racing industry are lipids known as the eicosanoids [12]. Eicosanoids are a large subclass predominately defined by the 20 carbon chain containing over 100 lipid mediators including prostaglandins, thromboxanes, leukotrienes, hydroxy fatty acids and lipoxins [12,15] with the majority derived from AA, an omega-6 fatty acid [16]. Eicosanoids are believed to act as inflammatory mediators since they have the ability to mimic inflammatory symptoms and decrease in the presence of anti-inflammatory drugs [8,14]. Disruption of eicosanoids can cause a range of inflammatory pathological conditions including asthma, chronic obstructive pulmonary disease, fevers, pain, a range of cardiovascular diseases and cancers [15]. Eicosanoids are synthesised at the site of injury in order to control and regulate the inflammatory response [7].

AA is released from membrane phospholipids through the activation of phospholipase A_2_ enzyme (PLA_2_) [14]. It can be further converted into other eicosanoids in the cascade (Figure 1). In a non-targeted sense, it should be possible to utilise the AA cascade to determine which lipids are being affected following drug administration. This cascade includes prostaglandin D_2_ (PGD_2_) [17], prostaglandin E_2_ (PGE_2_) [18], prostaglandin F_2α_ (PGF_2α_) [7], thromboxane B_2_ (TXB_2_) [19], 11-dehydro thromboxane B_2_ (11-Dehydro TXB_2_) [20], 6-keto prostaglandin F_1α_ (6-Keto PGF_1α_) [21], 15(S)-hydroxyeicosatetraenoic acid (15-HETE) [7], 5(S)-hydroxyeicosatetraenoic acid (5-HETE) [22], leukotriene B_4_ (LTB_4_) [23,24,25], leukotriene D_4_ (LTD_4_) [23,25] and leukotriene E_4_ (LTE_4_) [23,25]. The analogues of AA are also of interest including arachidonoyl ethanolamide (AEA) [26,27] and oleoyl ethanolamide (OEA) [26].

AEA is an endogenous cannabinoid ligand that has binding activity resulting in pharmacological effects of tetrahydrocannabinol (THC) such as euphoria and calmness [26,27]. In a variety of cells, cannabinoid agonists have caused an increase in the amount of AA [26]. This has been hypothesised to result from the combination of PLA_2_ and acyltransferase inhibition [26]. AA can also be converted into N-(4-hydroxyphenyl) arachidonylamide (AM404) which displays analgesic properties and the ability to lower body temperature [28,29]. AM404 is produced when acetaminophen is metabolised in the body to produce *p*-aminophenol, which is then conjugated with AA [29]. AM404 has been reported to inhibit the cyclooxygenase (COX) pathways leading to the decreased formation of PGE_2_, demonstrating effectiveness as a COX-2 enzyme inhibitor to reduce the production of prostaglandins by consumption of AA [29]. The COX and lipoxygenase (LOX) pathways are of particular interest for equine anti-doping due to their augmentation following anti-inflammatory treatments.

Prostaglandins are monocarboxylic acids with two side chains at carbon 7 and 8 attached to a central, five-membered ring [14]. Prostaglandins have oxygen-containing substituents in various positions in the molecule with the naming of the prostaglandins ranging from PGA to PGI depending on the basis of the substituents in the ring, and further sub-grouped into three series depending on the degree of unsaturation [14]. Prostaglandins are one of the key compounds in the generation of the inflammatory response due to an increase in concentration in inflamed tissue [30]. Prostaglandins are formed from AA being converted by the COX enzyme to prostaglandin G_2_ and H_2_ [17]. Prostaglandin endoperoxide-D-isomerase can convert PGH_2_ into a mixture of PGD_2_, PGE_2_ and PGF_2α_ [17]. PGH_2_ can also produce prostacyclin (PGI_2_) that may further metabolise to a more stable compound, 6-keto F_1α_ [19]. PGEs and PGIs have been identified to mimic the signs of inflammation caused by vasodilation and swelling due to an increase in vascular permeability [14].

Prostaglandins can also convert into thromboxane A_2_ (TXA_2_), an unstable intermediate in the production of TXB_2_, which further metabolises to form 11-dehydro TXB_2_ [20]. Thromboxanes are the major products of prostaglandin endoperoxides in platelets, lungs and the spleen [14]. The production of new platelets could potentially have a high capacity to further increase the synthesis of TXA_2,_ leading to increased amounts of TXB_2_ and 11-dehydro TXB_2_ [19]. The use of non-steroidal anti-inflammatory drugs (NSAIDs) results in PGI_2_ and thromboxane synthesis being inhibited [19].

Using the LOX pathways, AA converts into esterified hydroperoxyeicosatetraenoic acids (HPETEs) [31]. HPETEs are further reduced to their corresponding hydroxyeicosatetraenoic acids (HETEs) [7]. For example, using the 15-LOX enzyme, AA will metabolise into 15-HPETE, and then further metabolises to 15-HETE. Similarly, the 5-LOX enzyme converts AA to 5-HPETE and further to 5-HETE. The leukotrienes are an oxygenated metabolite of polyunsaturated fatty acids, but the initial formation is catalysed by lipoxygenases [23]. Leukotrienes are produced from AA using the 5-lipoxygenase (5-LOX) enzyme to produce Leukotriene A_4_ (LTA_4_), an unstable epoxide. LTA_4_ is hydrolysed to LTB_4_ or conjugated with glutathione to yield Leukotriene C_4_ (LTC_4_) and its metabolites LTD_4_ and LTE_4_. Leukotrienes are known for their strong vascular effect with the most effective being LTB_4_, in comparison to LTC_4_ and LTD_4_ [14,15]. In the presence of more leukocytes, the leukotrienes also have a role in the inflammatory process by increasing blood pressure [23].

## 4. Analytical Techniques for Lipidomics

### 4.1. Sample Preparation

With a wide range of analytical techniques applicable for doping analysis, optimised sample preparation methods have been developed [32]. There are typically two types of sample preparation strategies: protein precipitation (PP) which is monophasic; and lipid extractions which are biphasic [33]. For the latter, solid phase extraction (SPE) and liquid-liquid extraction (LLE) are commonly used [32,33]. LLE efficiency is improved if a mixture of solvents is used in comparison to only a single solvent [34]. SPE is a useful sample preparation method as it allows for the preconcentration of low sample volumes, which provides sensitive detection [35].

Chambers et al. [32] conducted a study comparing the various sample preparation methods including PP, SPE and LLE to optimise sample preparation for the analysis of eicosanoids. While PP was rapid and simple, it did not often provide a clean extract for analysis, however, the type of organic solvent used in the PP did influence the cleanliness of the extract. The MeOH extract contained 40% more potentially interfering phospholipids in comparison to the ACN extract. This result is consistent with the work of Bruce et al. [36] and Stojiljkovic et al. [5] showing that ACN or acetone was the best solvent for PP-based methods. ACN is more efficient in protein removal due to the higher dielectric constant and lower viscosity that it possesses [5]. Ammonium sulphate can also be used, however it has to be combined with SPE or a secondary LLE to ensure a cleaner extract before liquid-chromatography mass spectrometry (LC-MS) analysis [5]. SPE stationary phases are selected along similar principles to chromatographic column phases, with the size, polarity and charge of analyte considered. The pure cation exchange stationary phase has produced cleaner extracts, however, polymetric mixed-mode strong cation exchange can enable hydrophobic phospholipids containing two alkyl chains and lysophospholipids to be removed more efficiently. In comparison, LLE has been widely investigated and provides a cleaner method of sample preparation in comparison to PP [32]. The cleanliness of the extracts were comparable with cation-exchange mixed-mode SPE using three methods: a 3:1 ratio of methyl tert-butyl ether (MTBE) to human plasma, a 3:1 ratio of basified MTBE (5% NH_4_OH in MTBE) to human plasma and basified MTBE in a two-step extraction [32].

### 4.2. Analysis

A broad range of analytical techniques can be used to separate, detect and quantify eicosanoids including high performance liquid chromatography with ultraviolet detection (HPLC-UV) [37,38], enzyme-linked immunosorbent assay (ELISA) [39], nuclear magnetic resonance (NMR) [5], gas chromatography-mass spectrometry (GC-MS) [40] and LC-MS [15,39]. The main disadvantage of using HPLC-UV is the limited sensitivity and specificity of UV detection in complex biological matrices such as urine or plasma [15,37,38]. There is also the possibility of the lack of active chromophores in lipids that absorb UV light at the appropriate wavelengths [38]. ELISA is often limited to one analyte per assay and due to eicosanoids having a large number of isomers, there is the possibility of high cross-reactivity affecting quantification [39]. The use of NMR has its advantages as a non-destructive analysis involving minimal sample preparation, however the main disadvantage is the low sensitivity in comparison to GC-MS or LC-MS and the relatively large sample requirement [5]. GC-MS boasts high sensitivity and resolution of eicosanoids that have numerous isomers, however, complex sample preparation and derivatisation is usually required [40]. One of the most commonly used LC-MS instruments used is triple quadrupole mass spectrometry due to the high sensitivity and specificity obtained using multiple reaction monitoring acquisition for extremely low level lipids in the equine system [7]. With this high sensitivity, quanititation of eicosanoids has been possible which is further discussed in chapter 6 of this review.

The use of high-resolution mass spectrometry (HRMS) in metabolomics has enabled the indirect detection of substances that have a short half-life but long-lasting effects on biological systems [5]. Following this approach, studies have utilised LC-HRMS for lipidomics to investigate the potential to distinguish isomeric lipids [41]. Lipidomic studies are increasingly using LC-HRMS due to high sensitivity, mass accuracy, resolution and acquisition rates providing the ability to detect subtle differences in complex biological matrices such as plasma and urine [42,43,44,45]. The two types of HRMS instruments include the Quadrople Time-of-Flight mass spectrometry (QToF-MS) and the Orbitraps. These two HRMS instruments have the ability to achieve a mass accuracy of below 5 ppm for higher accuracy in the identification of compounds, with some instruments being adveritsed to have the ability to achieve less than 1 ppm difference [46,47]. Commerically available QToF instruments have resolving power between 35,000 and 70,000 full width at half maximum (FWHM) whilst comparatively, the Orbitraps have over 1,000,000 FWHM [47,48,49]. Whilst the Orbitraps do outperform the QToFs in mass resolving power, the disadvantage would be the longer accumulation times, which result in the Orbitraps being less suitable for accurate quanitification of compounds with narrow LC peaks [46]. Therefore, the identification of unknown lipids using LC-HRMS is slightly difficult due to the isomeric and isobaric nature of lipids, but the accuracy of these instruments has the potential to narrow the search to a particular class of lipids rather than the individual compound. The use of LC-HRMS allows for a complementary targeted/non-targeted approach to detect new and emerging drugs that could potentially be used as doping agents, and the metabolic signature that can result from such [5].

### 4.3. Data Acquisition Modes

Non-targeted lipidomics workflows allow for the generation of new hypotheses for supporting evidence in biological interpretations or for complementary ‘omics’ data. Targeted lipidomics is often used for the validation of biomarkers following the untargeted discovery phase or whilst trying to measure certain lipids that are associated with disease or are present in low abundance [50].

Two common modes of data acquisition for LC-HRMS analyses supporting lipidomic workflows are data dependent acquisition (DDA) and data independent acquisition (DIA) [47]. DDA utilises precursor ions that have exceeded an abundance threshold, predefined isotopic pattern and the presence of diagnostic product ions [47,48]. These ions, once detected, facilitate the acquisition of MS2 data, however, this is limited to ions of relatively high intensity [51]. There is also the possibility of precursor ions and relevant analytes being missed [52]. DDA is well suited to targeted screening, however reproducibility and variation in spectral databases often results in unknown metabolites being difficult to interpret [47].

Comparatively, DIA or sequential window acquisition of all theoretical fragmentation ion (SWATH) allows for simultaneous screening of all precursor ions and their fragmentation pattern within a specified mass to charge (*m/z*) window regardless of intensity [47,48,49,51,53,54]. The use of DIA for lipidomics was demonstrated after its potential was seen in proteomics [50]. DIA has been shown to be an effective means to produce a larger number of quantifiable results in short time periods with fewer errors relating to reproducibility across replicate analyses [55]. The freedom of ion choices reduces the need for data reacquisition due to unexpected ion formation or unrecognisable compounds [52]. Therefore, with the ability to screen multiple precursors, it is often used for untargeted screening. The major disadvantage of DIA is the potential for the incorrect precursor ion being identified as there is no criteria for ions to undergo dissociation [43]. It is also possible to obtain a mixture of (i.e., chimeric) spectra due to the wide scanning range of the *m/z* windows [43]. Therefore, in lipidomics, with a large number of isomers and levels of unsaturation, the ability to separate between lipid classes is more difficult [52]. Class separation for lipids is generally limited to DDA workflows where the ability to isolate unit mass would be more appropriate [52]. At the time of writing this review, the authors have identified a gap in the research where there is a lack of research focusing on the use of DIA on eicosanoids, specifically. There is research completed on lipids as an entirety focusing on whole classes but lacking on eicosanoids due to the vast number of lipid isomers and levels of unsaturation [52,56]. Another limitation of DIA is the possibility of isomeric or isobaric compounds being indistinguishable due to co-elution with an increased possibility of incorrect compound assignment as these compounds can act as interferences [43]. There is also the lack of spectral databases available for non-targeted screening as most databases utilise DDA [48].

## 5. Statistical Analysis

A comprehensive lipidomics study of molecules in any biological system can result in a large amount of data generated. There are two common approaches to process lipidomic data: chemometric and quantitative [57]. Chemometric analysis is performed on spectral patterns or signal intensity data to identify any lipids through non-targeted screening. Chemometric analysis can be automated, however, there is the uncertainty of strict sample uniformity and repetition. Quantitative analysis in comparison is more applicable to biological studies as it allows for the identification of all lipids proceeding to the analysis of the lipid data.

The information collated from quantitative studies are either univariate or multivariate depending on the study being conducted. Saccenti et al. discuss the advantages and disadvantages of univariate and multivariate analyses of metabolomics data [58]. Univariate analysis is performed when only one variable is analysed, and it includes methods to test different sets of samples such as ANOVA or *t*-tests. There are, however, disadvantages with univariate analysis with higher possibilities of equivocal findings due to the requirement for multiple testing corrections. Univariate methods are usually the preferred choice of statistical analysis for biologists due to the ease of interpretation the analysis provides. In comparison, multivariate analysis consists of data that have two or more variables. There are many tools available for multivariate analysis including correlation analysis and simple linear regression which can analyse a data set that contains several hundred or more variables simultaneously, thereby eliminating the need for multiple univariate methods. Multivariate statistical analysis provides advantages that univariate may not, including the ability of independent variables to complement each other for the prediction of a dependent variable. Lipidomic studies are often multivariate in nature due to a large number of compounds investigated, however, it is also common practice for both univariate and multivariate approaches to be used in a complementary manner [58].

Data are often analysed using both unsupervised and supervised strategies [42]. Unsupervised studies allow the discovery of groups or trends in the data with the most common tool being principal component analysis (PCA). Supervised methods are then often used to discover new biomarkers with partial least squares (PLS) being one of the most common. PLS is a multidimensional method that utilises a data matrix containing independent variables and relates it to dependent variables [42]. As biomarker research usually involves multiple approaches, it is recommended to combine both unsupervised and supervised methods [58]. Various statistical software programs have been utilised to implement supervised and unsupervised methods. One such software program is the Mass Spectrometry-Data Independent Analysis software version 4 (MS-DIAL 4), where a non-targeted lipidomics platform has been utilised to provide a comprehensive lipid database [59]. This contains retention times, mass-to-charge ratios, isotopic ions, adduct information and other mass spectral information for the possible identification of unknowns and lipid pathways [59]. However, there are many limitations that arise with trying to determine the putative structure of an unknown given the limited information provided from statistical analysis [1,60,61]. Confirmations of putative biomarkers is difficult as comparisons need to be made with authentic reference standards if possible, if not, there is the possibility of custom synthesis, but these are highly cost inefficient and may incur timely delays for a highly purified sample [60]. The use of these statistical analyses will help provide the information required to further improve current routine testing using non-targeted workflows to monitor for new biomarkers indicative of doping.

## 6. Studies Monitoring Eicosanoids for Equine Anti-Doping Screening

The administration of corticosteroids and NSAIDs to alleviate pain and inflammation is known to affect the amount of AA, with the COX and LOX pathways subsequently reducing the concentration of eicosanoids present [62,63,64,65,66,67]. The use of approved corticosteroids and NSAIDs, whilst being legitimate therapeutics for racehorses out-of-competition, are controlled for race day competition. Therefore, the monitoring of terminal compounds in the AA cascade may prove beneficial for the detection of these prohibited substances. Glucocorticoids are potent anti-inflammatory agents that increase the tolerance for pain which may allow horses to compete under conditions which could compromise the health of the individual horse and safety of riders. Early studies on eicosanoids were performed using ELISA and high-performance liquid chromatography, however, these techniques are known to lack specificity [7]. Currently, advanced technology of hyphenated mass spectrometry such as triple quadrupole mass spectrometry and QToF instruments are being used. These instruments have enabled an enhanced scope of eicosanoid monitoring [7] for targeted screening, however, the use of non-targeted methods have not been extensively explored. Therefore, there is potential in future studies for untargeted methods to provide evidence of an exogenous administration.

### 6.1. Lipid Screening in Equine Samples

Jackson et al. [18] was one of the first studies to explore the use of lipids (specifically PGE_2_) in lipopolysaccharide (LPS)-stimulated blood for a comprehensive NSAID screening tool using ELISA. However, there was a moderate degree of variability between the different horses tested due to the LPS-stimulation. Even so, PGE_2_ production was less than 50% between 8- and 12-h post-administration. Therefore, it was proposed that an LPS-stimulated plasma concentration of PGE_2_ of less than 500 pg/mL could potentially indicate the administration of an NSAID given the terminal position of PGE_2_ in the cascade.

Nolazco et al. [11] studied the detection of lipids in racehorses before and after supramaximal exercise utilising an LC-QToF-MS. A non-targeted approach discovered 933 lipids present in the plasma of which 130 were known based on library matches. One-tenth (i.e., 13 out of 130) were deemed statistically different compared to baseline concentrations. From these, three unsaturated fatty acids and six phospholipids displayed an increase in signal intensity, whilst four saturated fatty acids (including n-eicosanoid acid) and five triacylglycerols had a decrease in signal intensity. Various hypotheses were proposed to explain the results which included that during exercise, lipolysis causes the ratio of unsaturated to saturated fatty acids for triacylglycerols contained in adipose tissue to be higher than non-esterified fatty acids. Another possibility is that during exercise, the use of saturated fatty acids as an energy source is preferred over unsaturated fatty acids, causing an increase in concentration. Whilst this study provided information about the lipidome changes that occur during exercise in racehorses, limitations include the use of a treadmill rather than a racetrack and the small sample size due to the availability of horses. This study demonstrated that non-targeted lipid profiling aimed at detecting anti-inflammatory drug use needs to account for the effects of exercise.

### 6.2. Lipid Inflammatory Markers and Their Effect from a Corticosteroid Administration

Corticosteroids including dexamethasone, triamcinolone acetonide (TACA) and flumetasone are commonly used throughout the racing industry to alleviate symptoms of inflammation and prevent further tissue damage caused by inflammatory markers [64].

Mangal et al. [7] and Knych et al. [64] explored the use of dexamethasone on the effects of inflammatory mediators. Mangal explored 25 eicosanoids in equine plasma using stable isotope dilution reversed-phased LC-MS, whilst Knych utilised triple quadrupole mass spectrometry to monitor six eicosanoids. Both Mangal and Knych employed calcium ionophore (CI) A23187 or LPS-simulation of whole blood, whilst Mangal also used AA added exogenously to determine whether extra enzymatic activity would increase the production of eicosanoids. The use of CI allows for any free arachidonic acid to be released from surrounding cellular membranes to produce eicosanoids through either COX or LOX enzyme activity [62,63]. The use of LPS-stimulation allows for the modelling of COX enzymes for the production of prostaglandins and thromboxanes from arachidonic acid [30,62,63]. Mangal concluded from the 25 monitored eicosanoids, 9 resulted in reduced levels [7]. Comparatively, Knych concluded that whilst dexamethasone does not have a direct effect on the COX-1 enzyme, the effect could be seen on the COX-2 enzyme due to LPS stimulation [64]. Knych also observed that the LOX pathways were affected, specifically 5-LOX, due to the nuclear-factor (NF) kappa beta enzyme which is a known inducer of the 5-LOX enzyme [64]. From CI-stimulation, significant down-regulation was observed for four eicosanoids, while LPS-stimulation showed five eicosanoids to be significantly down-regulated [64].

TACA is a commonly used, long-lasting and potent glucocorticoid that is known to bind to glucocorticoid responsive genes. This can result in increased concentrations of anti-inflammatory mediators whilst decreasing the inflammatory markers, as demonstrated by another study conducted by Mangal et al. using a triple quadrupole mass spectrometer [62]. CI and LPS stimulation were again used to explore the inflammatory mediators in equine plasma. For CI-stimulation, a significant increase for two inflammatory mediators was observed but only one proceeded to show a significant decrease. Comparatively, for LPS-stimulation, four eicosanoids showed significant reduction in concentration, then a significant increase in concentration prior to a return to basal levels. This study concluded that major products expressed through the COX-2 enzymatic activity are down-regulated under the influence of an external glucocorticoid suggesting inhibition of the COX-2 enzyme [62]. It is worth noting that there is speculation that PGE_2_ has the potential to have a binary role in inflammation given that it is produced using the COX-1 enzyme as an inflammatory mediator, however delayed production of PGE_2_ by the m-PGES-1 enzyme, coupled to COX-2 gives PGE_2_ its anti-inflammatory properties [62,68].

Flumetasone is a potent corticosteroid often used in the treatment of performance related injuries associated with strenuous exercise [69]. Knych et al. explored the effects of Flumetasone on the inflammatory markers under the stimulation of CI and LPS using a triple quadrupole mass spectrometer. The authors concluded that Flumetasone had a direct inhibitory effect on the COX and LOX enzymes, however, further studies will be required for the specific effect on the 15-LOX enzyme [69]. This was demonstrated with significant change for two eicosanoids in the 5-LOX enzymatic pathway following CI-stimulation [69]. Comparatively, utilising LPS-stimulation, six eicosanoids displayed a significant reduction indicating a direct effect on the COX enzyme [69].

### 6.3. Lipid Inflammatory Markers and Their Effect from an NSAID Administration

NSAIDs are known to predominately inhibit the COX-2 enzyme rather than COX-1 [65]. Cuniberti et al. utilised several NSAIDs, namely eltenac, Naproxen, tepoxalin, SC-560 and NS 398, to study their effects on both COX-1 and COX-2 enzymes using a chromatographic assay [65]. This study concluded that certain NSAIDs (e.g., SC-560, NS 398 and eltenac) can act as either COX-1 or COX-2 enzyme inhibitors whilst other NSAIDs (e.g., tepoxalin and Naproxen) can act as dual inhibitors [65]. This was seen with an in vitro model demonstrating 100% inhibition of three eicosanoids by all drugs whilst ex vivo models also showed signs of inhibition but not to the same extent.

Phenylbutazone (PBZ) is one of the most commonly used NSAIDs in equine medicine for the treatment of training and performance related injuries [63]. The use of PBZ is highly regulated due to the potential to mask injuries during pre-race (i.e., fitness-to-race) examinations [63]. Knych et al. explored how PBZ affected the biomarkers of inflammation following intravenous (IV) and oral administration whilst also utilising stimulation with CI and LPS [63]. The technique used for this study was a triple quadrupole mass spectrometer. The duration of monitored effects of PBZ exceeded the direct detection of PBZ in plasma [63]. Utilising CI-stimulation, there was a significant change for two eicosanoids whilst LPS-simulation showed four eicosanoids to significantly change indicating PBZ is an effective inhibitor of the COX-1 enzyme [63].

### 6.4. Lipid Inflammatory Markers and Their Effect from a Cannabidiol Administration

Cannabidiol (CBD), a known antidote for inflammation in humans, has recently been of concern with growing interest from horse trainers considering the use of hemp-based supplements [70]. The effects of CBD on the equine system have not been extensively studied, however, it is believed that there are multiple mechanisms that CBD follows which potentially impact the endocannabinoid pathway including inflammatory, analgesia and stress responses [70]. There is a hypothesis that CBD inhibits COX and LOX enzymes which decreases the amount of pro-inflammatory eicosanoids that are produced during the inflammatory response [70]. This hypothesis follows a similar mechanism for NSAIDs which have been demonstrated to affect the inflammatory pathways. Ryan et al. recently evaluated the anti-inflammatory effects of CBD with results indicating that both COX-1 and COX-2 enzymes were being stimulated [70] utilising an Agilent 1260 chromatography system coupled to a triple quadrupole mass spectrometer. Whole blood samples were stimulated using either CI or LPS to monitor the inflammatory mediators. During LPS-stimulation, four eicosanoids were observed to have significant change indicating the effects of both COX-1 and COX-2 enzymes, however, there is also evidence of CBD being dose dependent with a higher dose resulting in stronger enzyme inhibition [70]. During CI-stimulation, three eicosanoids following the LOX pathways were affected [70]. The major limitation with this study is that the endogenous levels of these inflammatory mediators are not known since the study used an ex vivo model of inflammation.

## 7. Potential for Lipidomics to Contribute Longitudinal Assessments for Equine Anti-Doping

The primary focus of anti-doping analyses is the detection and quantification of exogenous drugs or substances in biological matrices such as urine and plasma [4]. However, issues such as the clear distinction between endogenous and exogenous origins of naturally occurring substances and varying individual levels of biomarkers have been evident for decades. Therefore, the need for a longitudinal perspective is necessary given that it can also be an indicator of pharmaceutical manipulation. The equine biological passport (EBP) can provide a basis for complementing targeted analysis (detecting prohibited substances) with the non-targeted monitoring of biomarkers, which may be present after the original substance has been eliminated, metabolised or excreted [71]. The establishment of individual reference limits has been useful in monitoring levels of endogenous biomarkers to determine their respective basal levels, which may differ from the general population. This form of monitoring allows for an indirect approach to monitor biomarkers in order to potentially increase the detection window of administered drugs. The EBP has dual purposes: allowing for the possible detection of novel doping agents together with directing the resources of racing authorities to participants that may be engaging in prohibited practices [71]. It also allows for the monitoring of substances considered relevant to integrity but which do not have an agreed international threshold that exists to control specific misuse [71]. The EBP can provide a deterrent to the use of prohibited substances, since it has the potential to measure the biological effect of an administered substance for longer periods than the presence of the substance itself [71]. As outlined in this review, a lipidomic component may provide a useful contribution to an EBP. For future work, it would be ideal to include lipids into the EBP based off of lipids that were identified from complementary targeted and untargeted research. To the authors’ knowledge at the time of writing this review, this potential has not been presented before highlighting the novelty of the passport and the use of lipids throughout the passport.

## 8. Conclusions

In conclusion, the improved use of biomarkers for a complementary targeted/non-targeted approach has been explored to potentially extend the time of detection for an administered substance. The current issues involving the use of targeted metabolomics and lipidomics such as short detection windows and matrix effects require investigation of non-targeted screening strategies. Lipidomics is an expanding field within the broader area of metabolomics focusing on the lipidome in any system with common polyunsaturated fatty acids being a starting point in the potential for a non-targeted approach. Whilst current research still utilises targeted methods to monitor biomarkers as seen in the various equine studies (including corticosteroids, NSAIDs and cannabidiol), there is considerable potential for non-targeted studies in the future. This can be attributed to improved sample preparation in conjunction to the increased use of high-resolution accurate mass spectrometry for non-targeted methods. The use of DIA and DDA to monitor biomarkers not already currently monitored, in addition to more sophisticated statistical analyses, will likely realise the potential for non-targeted lipidomic studies in the equine racing industry.

## Figures and Tables

**Figure 1 molecules-28-00312-f001:**
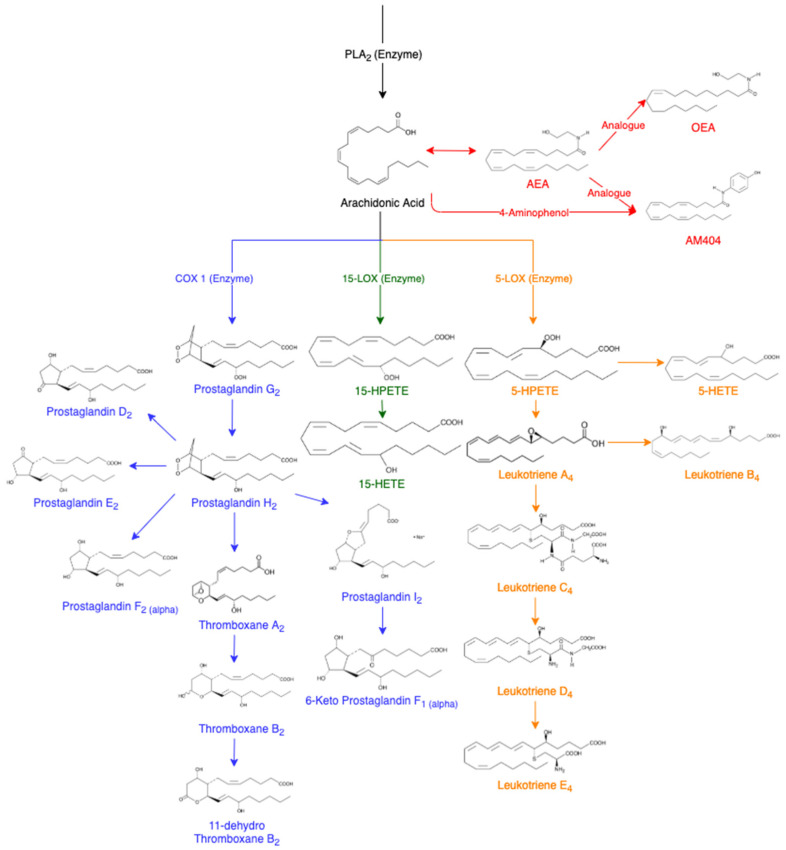
Arachidonic Acid Cascade, adapted from various sources [7,8,14,17,18,19,20,21,22,23,24,25,26,27,28,29]. Abbreviations of eicosanoids: arachidonoyl ethanolamide (AEA), oleoyl ethanolamide (OEA), prostaglandin (PG), thromboxane (Tx), hydroperoxyeicosatetraenoic acids (HPETES), hydroxyeicosatetraenoic acid (HETE) and leukotriene (LT).

## Data Availability

Not Applicable.

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
