# Peer review of "Towards Non-Targeted Screening of Lipid Biomarkers for Improved Equine Anti-Doping"

_molecules, 2022, doi:10.3390/molecules28010312_

Round 1
Reviewer 1 Report
The paper "Towards non-targeted screening of lipid biomarkers for improved equine anti-doping" presents the detection of prohibited substances based on lipidomic investigations.
- Originality/Novelty: The paper is novel as it studies the latest practices to monitor biomarkers for equine, namely corticosteroids, non-steroidal anti-inflammatory drugs and cannabidiol.
- Significance: The results of the research are interpreted properly, the fine tuning of model parameters being performed.
- Quality of Presentation: The article is written appropriately, respecting the logical succession of sections. Data and analyses are presented graphically. The results were outlined using high standards, the advantages of the biomarkers monitoring being very clear.
In the abstract it is mentioned that the review will also explore the techniques used to prepare and analyse samples, but this is actually done in the paper already, so it can be stated that the review explored the techniques. It is just the tense of the verb which needs to be changed.
Please leave a space after the word and the reference.
Please avoid the usage of pronouns like "we" or "I" in a journal paper.
The conclusions section has to include further details about future work.
How can an even better accuracy be obtained?
- Scientific Soundness: The paper offers enough details to allow the reproduction of the results, but it needs to have a higher scientific value.
- Interest to the Readers: The content of the article would surely interest the readers of the Molecules journal, and not only them.
- Overall Merit: The findings and their implications should be discussed in the broadest context possible and limitations of the work should also be further highlighted. Please present the novelty as compared to the previously published papers.
- English Level: The level of English language is advanced. Through the entire paper, the language was appropriate and understandable, being easy to follow the flow from the beginning.
Reviewer 2 Report
This review summarizes current applications of lipidomic approach to equine anti-doping strategy, especially focusing on detection and quantification of eicosanoids such as prostaglandins, leikotriens and HETES. This is a comprehensive review and would help people who are unfamiliar to this research field. On the other hand, addition of more description would be desired for better understanding of this article. Details are as follows.
1, In Fig. 1, structure of each molecule should be added beside the compound names.
2. In Chapter 4.2, "HRMS" and "LC-HRMS" were shown to be used for lipidomics. These words show that they are high resolution, but do not show how much high the resolutions is, and what types of mass spectrometers they are. Please specify the types of machines and their spec more in detail.
3, In Chapter 4. it should be indicated which paper showed that eicosanoids could be detected by DIA, and which kind of eicosanoids were detected there, because this review is focused on eicosanoids.
4. In Chapter 4, it is necessary to mention about the methods for "quantitation" of lipid by mass spectrometry. In addition, Chapter 4 should also have brief explanation about triple quadrupole mass spectrometry, since it was mentioned in Chapter 6.2.
5. In Chapter 6, it was mentioned that "hyphenated mass spectrometry" has been used for eicosanoid monitoring. What is this MS ?
6. In Chapter 6, what kind of mass spectrometric strategy was used in the study of ref. 11, and what kind of eicosanoids detected?
7. Chapter 6.3, what kind of mass spectrometric strategy was used for the study of PBZ ?
